# Genome-Wide Association Mapping for Heat and Drought Adaptive Traits in Pea

**DOI:** 10.3390/genes12121897

**Published:** 2021-11-26

**Authors:** Endale G. Tafesse, Krishna K. Gali, V. B. Reddy Lachagari, Rosalind Bueckert, Thomas D. Warkentin

**Affiliations:** 1Department of Plant Sciences, College of Agriculture and Bio-Resources, University of Saskatchewan, Saskatoon, SK S7N5A8, Canada; endale.tafesse@usask.ca (E.G.T.); kishore.gali@usask.ca (K.K.G.); rosalind.bueckert@usask.ca (R.B.); 2AgriGenome Labs Pvt. Ltd., Hyderabad 500078, India; vb.reddy@aggenome.com

**Keywords:** pea, heat, drought, stress, genome-wide association study, genotyping-by-sequencing, marker–trait association

## Abstract

Heat and drought, individually or in combination, limit pea productivity. Fortunately, substantial genetic diversity exists in pea germplasm for traits related to abiotic stress resistance. Understanding the genetic basis of resistance could accelerate the development of stress-adaptive cultivars. We conducted a genome-wide association study (GWAS) in pea on six stress-adaptive traits with the aim to detect the genetic regions controlling these traits. One hundred and thirty-five genetically diverse pea accessions were phenotyped in field studies across three or five environments under stress and control conditions. To determine marker trait associations (MTAs), a total of 16,877 valuable single nucleotide polymorphisms (SNPs) were used in association analysis. Association mapping detected 15 MTAs that were significantly (*p* ≤ 0.0005) associated with the six stress-adaptive traits averaged across all environments and consistent in multiple individual environments. The identified MTAs were four for lamina wax, three for petiole wax, three for stem thickness, two for the flowering duration, one for the normalized difference vegetation index (NDVI), and two for the normalized pigment and chlorophyll index (NPCI). Sixteen candidate genes were identified within a 15 kb distance from either side of the markers. The detected MTAs and candidate genes have prospective use towards selecting stress-hardy pea cultivars in marker-assisted selection.

## 1. Introduction

Pea (*Pisum sativum* L., 2n = 14) is among the world’s most cultivated pulse crops, and its economic value is mainly derived from its nutritious seed that is high in protein, slow-digestible starch, essential minerals, dietary fiber, while being low in fat [1,2]. However, like many crops, pea is prone to various environmental stresses, predominantly to drought and heat, that can lead to a significant yield loss [3,4]. Yield loss primarily arises from a shortened life cycle, reduced pollination and seed set, and the abortion of flowers and pods [5,6]. Unfortunately, several climate models predict future crop production will be increasingly challenging due to global warming, extreme heat, and the escalating frequency of severe drought in several places across the world [7,8]. As such, new crop cultivars need to be better adapted to stressful conditions. 

To adapt and succeed in stressful microenvironments, plants have developed sophisticated mechanisms that may involve morphological, physiological, and biochemical alterations [9,10]. The mechanisms can generally be classified into long-term evolutionary modifications to morpho-anatomical architecture and phenological alterations, or immediate stress aversion, such as through reflecting excess radiation and energy dissipation, and stomatal closure [4,11]. Epicuticular waxes in plant canopies form the primary interaction between the canopy and the environment and play a vital role as a protective layer of the canopy from environmental stresses such as excessive radiation and heat. As a drought-tolerance trait, leaf wax minimizes the excess loss of water through stomatal and non-stomatal transpiration [12]. Surface wax has been extensively studied as a stress resistance trait in multiple crops including pea, sorghum and wheat [13,14,15,16]. 

With regard to plant architecture and canopy types, Tafesse et al. [4] reported lodging resistant upright pea cultivars with semi-leafless leaf types to be stress hardy. A thicker stem may maintain more water plants and enhance leaf water potential. Klepper et al. [17] reported a positive association between stem thickness and plant water status in cotton, and thus would contribute to both heat and drought resistance. 

Flowering duration is highly dependent on rainfall and temperature. One of the direct effects of heat stress is shortening the flowering duration and accelerating plant senescence [5]. In a recent study using 24 diverse pea cultivars, Tafesse [18] reported a range of 15 to 28 days in flowering duration. Cultivars with a longer flowering duration resist environmental stresses by being able to compensate yield by prolonging the plant growth and development duration in the field [19]. Vegetative indices have been widely used as indirect methods to evaluate traits associated with plant growth, pigment composition and abundance, and water content [20,21,22]. For example, the normalized difference vegetation index (NDVI) has been widely used as a proxy to determine plant vigor, yield, drought stress, and overall vegetation health [23,24,25]. The normalized pigment and chlorophyll index (NPCI) is considered to be a direct indicator of chlorophyll degradation, plant senescence, and the degree of stress in plants [21,26]. 

Genetic improvement of pea for stress resistance is a promising approach to develop cultivars that would grow and yield well under stress conditions. Pea germplasm has a wide range of diversity in morpho-anatomical, biochemical, and physiological characteristics that might be associated with traits of stress resistance [3,4,6]. The genome-wide association study (GWAS) has emerged as a powerful tool for scanning genetic regions that control various traits based on the naturally existing genetic diversity accumulated over several generations [27,28,29]. Linkage disequilibrium, the association of alleles at different loci, is the foundation of association mapping that provides a greater mapping resolution of quantitative trait loci (QTL) [30]. The emergence of high-throughput next-generation sequencing (NGS) technologies at an affordable price has made use of genome-wide SNPs ideal for genetic diversity studies and linkage disequilibrium estimation in numerous crops, including pea [29,31,32].

In pea, association mapping has been used to reveal the genetic regions controlling several traits including disease resistance [33], yield and yield components, seed quality [29], seed mineral concentrations [34], and others. However, only a few studies have been undertaken to uncover the genetic basis of resistance to environmental stresses. Drought and heat resistance are complex traits governed by several genes and environmental interactions. The three objectives of the current study were to evaluate the genotype by environment interaction for lamina and petiole epicuticular waxes, stem thickness, flowering duration, and vegetative indices connected with stress response in pea; investigate the genetic variation of stress-adaptive traits present in a GWAS panel composed of 135 accessions; and detect markers and candidate genes associated with these traits.

## 2. Materials and Methods

### 2.1. Plant Materials

The GWAS panel composed of 135 genetically diverse pea accessions previously assembled by Gali et al. [29] was used in this study. These diverse accessions originated from 23 pulse crop breeding programs across the world. The accessions primarily consisted of cultivars released over the past 50 years, and the specific countries of origin for each accession were described by Gali et al. [29]. 

### 2.2. Field Trials and Plant Growth Conditions

For phenotypic evaluation, the accessions were grown as follows: Two years (2016–2017) at Rosthern (52°66′ N, 106°33′ W) and three years (2015–2017) at Saskatoon (52°12′ N, 106°63′ W), Saskatchewan, Canada. The year–location combination produced five environments (2015 Saskatoon, 2016 Rosthern, 2016 Saskatoon, 2017 Rosthern, and 2017 Saskatoon) for phenotypic evaluation. The trials at each environment were laid in a randomized complete block design with two replications. The plot size was 1.37 m width × 3.66 m length in three rows seeding, and seeding density (100 seeds m^−2^, aiming 80–85 plants m^−2^ on 0.25 m row spacing) was used. Fertilization and weed management were achieved by best agronomic practices recommended for pea production in Saskatchewan, as presented by Tafesse et al. [4]. Weather data were collected from weather stations (Coastal Environmental Systems, Seattle, WA, USA) installed at each site, and obtained from the Environment Canada database (https://climate.weather.gc.ca (accessed on 12 January 2020)) for the nearest stations for any missing weather information from our stations, as described by Tafesse et al. [35]. 

Based on weather parameters including the mean daily maximum and minimum temperature, mean daily temperature, the number of days when the daily maximum temperature was greater than 28 °C (a threshold temperature that leads to a significant yield loss in pea; Bueckert et al., 2015), and total monthly precipitation, the five environments were grouped into stress and control conditions. Accordingly, Saskatoon 2015 and 2017 had greater daily maximum air temperatures > 28 °C, more warm days, limited rainfall, and drier conditions during reproduction, and were therefore classified as stress environments. The remaining three environments (2016 Rosthern, 2016 Saskatoon, and 2017 Rosthern) were mostly ambient and classified as control environments [35]. 

### 2.3. Phenotyping

For wax determination, one representative fully expanded leaf from either the second or third node on the main stem, counting down from the top of the plant, was cut from each plot on measurement days, put in a plastic bag, and transferred to the laboratory. These leaf samples were taken twice, at early flowering, and the full pod stage about 20 days later. To determine the projected surface area (cm^2^), the leaves were sorted into lamina and petiole parts, and each part was first scanned using winRHIZO (Regent Instruments Inc, Quebec City, Canada). Wax was extracted and quantified according to methods used on pea [14] developed by Ebercon et al. [13]. Briefly, bulk wax was removed from each sample by rinsing the lamina or petiole sample in 10 mL chloroform for 15 s at room temperature. After removing the plant tissue, the chloroform remaining in the tubes was evaporated using a water bath at 70 °C, which left wax residue on the walls and bottoms of the tubes. Then 5 mL of acidic K_2_Cr_2_O_7_ (20 g K_2_Cr_2_O_7_ per L of H_2_SO_4_) was added to each tube that contains the wax and boiled in a water bath at 100 °C, for 30 minutes. After cooling, 5 mL distilled water was added to each tube, the mixture was vortexed, and spectral absorbance was recorded at 590 nm using an Agilent 8453 diode array spectrophotometer with 1.6 ± 0.5 nm resolution, equipped with Chem Station software for UV-visible spectroscopy (Agilent Technologies, Santa Clara, California, USA). Wax concentrations were determined using a prediction equation that was developed from a linear (R^2^ > 0.98) relationship of a series of known beeswax concentrations (0, 0.66, 1.42, 3.33, 10 µg mL^−1^) using the same reagents. 

The stem thickness of the internode between the second and third nodes, counting down from the top, was measured at physiological maturity using a digital caliper (Model H7352, accuracy ± 0.02 mm). Flowering duration, the number of days elapsed from when 50% of the plants in a plot had started flowering to when 50% of the plants had terminated flowering on the main stem, was determined by taking flowering notes twice a week during the reproductive growth stage. 

Spectral reflectance measurements on stipules were taken once per week four times per plot during the reproductive stage for each of the five environments using a portable spectroradiometer (Model PSR-1100F, Spectral Evolution Inc, Lawrence, MA, USA). This instrument is capable of measuring hyperspectral reflectance in a light spectrum with a range of 320–1126 nm, at a 1.6 nm reading interval. From the spectral measurements, the normalized difference vegetation index (NDVI) and normalized pigment and vegetation index (NPCI) were calculated as follows: NDVI = (R_760_ – R_680_)/(R_760_ + R_680_)(1)
NPCI = (R_680_ − R_430_)/(R_680_ + R_430_)(2)
where R is the reflectance percentage and the numbers in subscript represent the specific wavelengths used [21].

### 2.4. Phenotypic Data Analysis

Prior to performing analysis of variance (ANOVA), the normal distribution of residuals and homogeneity of variances were tested using Shapiro–Wilk and Levene tests, respectively. Then, the variance components of the genotype, environment, the G × E interaction, replication, and the residual were determined using the generalized linear model (GLM) of SAS (Version 9.4, SAS Institute, Cary, NC, USA) and considering all factors as random effects. Broad-sense heritability (H2) was computed as:(3)$$H^{2}=\sigma^{2}g/(\sigma^{2}g+\frac{\sigma^{2}\mathrm{ge}}{n}+\frac{\sigma^{2}e}{nb})$$
where $\sigma^{2}g$ is the genetic variance, $\sigma^{2}\mathrm{ge}$ is the interaction variance between accessions and environments, $\sigma^{2}e$ is the error variance, n is the number of environments, and *b* is the number of replications in each experiment [36]. We then performed an ANOVA for the lamina wax, petiole wax, stem thickness, flowering duration, NDVI, and NPCI separately using the Mixed procedure of SAS (Version 9.4, SAS Institute, Cary, NC, USA). Genotype, environment, and the G × E interaction were considered as fixed terms while replication was assigned as a random term. Finally, principal component analysis (PCA) was employed with the multivariate function of Minitab (Version 21, Minitab LLC, State College, PA, USA) using the means of traits to determine the overall associations of traits among themselves under the stress and control environments. 

### 2.5. Genotyping

The genotyping data of the GWAS panel, previously reported by Gali et al. [29], were used in the current study. The 135 accessions were originally genotyped by the genotyping by-sequencing (GBS) method described by Elshire et al. [32]. The pea genome sequence by Kreplak et al. [37] was used as a reference for SNP allele calling. A set of 16,877 useful markers, selected based on a minimum read depth of five and minimum allele frequency of 0.05, was used for the analysis of the population structure [29]. The sequence information of these SNP markers is available at (https://www.ebi.ac.uk/ena/browser/view/PRJEB35147 (accessed on 15 April 2021)). The designation of a marker was made according to the pea chromosome number, linkage group number, and base pair position of the SNP. SNPs aligned to non-chromosomal scaffolds were labeled based on their respective scaffold number and base pair position.

### 2.6. Association Mapping

To test the association between SNP markers and the six traits used in the current study, GAPIT (Genome Association and Prediction Integrated Tool)—R package [38] software was used for individual environments. To reduce errors related to environmental variation, best linear unbiased predictors (BLUPs) of each trait of three or five environments were calculated using the ‘Ime4’ package of R3.6.1 software (www.r-project.org (accessed on 17 October 2021)). We conducted an association analysis for each trait using different models, including mixed linear model (MLM), multiple mixed linear model (MLMM), SUPER, and FarmCPU of the GAPIT program. For the association analysis, Q values were determined from structure analysis and K (kinship coefficient matrix) values were calculated by GAPIT and identity-by-state (IBS) methods. Based on the Q-Q plots, we selected the MLMM model to report the marker–trait associations. Marker-trait associations were declared based on both the P-value (*p* ≤ 0.001) and repeated occurrence of the association in multiple environments. Manhattan plots were produced to display the P-values distribution for the SNP markers. We used a *p* ≤ 0.001 significance level to assert the marker–trait association. Significant SNPs associated with the six traits were identified and the presence of the significant markers for each environment was observed in the quantile–quantile (Q-Q) plots.

### 2.7. Identification of Candidate Genes

Based on the physical position of selected SNP markers, the candidate genes present within a 15 kb distance on either side of the SNP were retrieved from the genome sequence of pea [37]. The 15 kb distance was used based on the LD decay observed by Gali et al. [29] in the GWAS population used in this study. The identified genes were searched against the NCBI-nr protein database using the BLAST program. The gene ontology (GO) terms associated with the genes were determined using BLAST2GO software [39].

## 3. Results

### 3.1. Phenotypic Distributions

Analysis of variance indicated that both the environment and genotype had significant (*p* < 0.001) effects on all traits (Table 1). In contrast, the genotype-by-environment interaction effect was non-significant for all traits except for flowering duration (Table 1). Based on the variance component analysis, the percentage total variance of each of the sources of variations is presented in Table 1. Out of the total variation for lamina wax, the contributions of genotype, environment, and residual error, respectively, were 30.0%, 41.5%, and 21.1%. For petiole wax, 39.8% of the variation was explained by the genotype, 36.3% by the environment, and 28.1% by the residual error. For stem thickness, 34.7% of the total variation was explained by the genotype, 33.9% by the environment, and 26.9% by the residual error. For the flowering duration, the genotype and environment explained 36.6% and 44.6% of the total variation, respectively, whereas the genotype-by-environment interaction and residual error, respectively, had contributions of 6.6% and 12.5% to the total variation. For both NDVI and NPCI, the genotype explained 45% and 61.4%, the environment explained 16.7% and 3%, and the residual errors explained 38.4% and 34.9% of the total variation, respectively. Broad-sense heritability indicates the proportion of phenotypic variation due to genetic factors that may be additive, dominant/recessive, and epistasis effects. The heritability value of the six traits was moderately high, ranging from 0.70 to 0.82. 

Overall, the phenotypic variation of the six traits across the 135 accessions and three or five environments was substantial (Table 2; Figure 1). Lamina wax concentrations ranged from 5.4 to 66.8 µg cm^−2^ across the accessions in three environments (2015 Saskatoon, 2016 Rosthern, and 2016 Saskatoon. On average, 2015 Saskatoon (the stress environment) had a 71% higher lamina wax concentration than the mean of control environments. Similarly, the petiole wax concentration ranged from 18.2 to 140.1 µg cm^−2^ across the accessions in three environments, and on average, the stress environment had a 33% higher petiole wax concentration than the mean of the control environments (Table 2). Stem thickness, flowering duration, NDVI, and NPCI ranged from 2.42 to 4.81 mm, 12.7 to 38.9 days, 0.64 to 0.85, and 0.21 to 0.70, respectively, across the accessions in the five environments. On average, the control environments had 17.0%, 22.7%, and 2.2% higher stem thickness, flowering duration, and NDVI values, respectively, than the stress environments. In contrast, NPCI was greater under the stress conditions than under control conditions (Table 2; Figure 1).

A principal component analysis of mean values of each accession under stress and control conditions revealed the phenotypic variability among the accessions, the overall traits associations, and the accessions’ response to the growing conditions (Figure 2). PC 1 explained 43.6% of the phenotypic variance based mainly on stem thickness (ST), flowering duration (FD), and NDVI. PC 2 explained 27.2% of the phenotypic variance represented mainly by lamina wax (LWAX), petiole wax (PWAX), and NPCI (Figure 2). Lamina wax was situated in an opposing direction to flowering duration and NDVI demonstrating a significant negative correlation of lamina wax with flowering duration and NDVI. Likewise, petiole wax was positioned in the opposite direction of ST indicating their inverse correlation. Traits positioned in the same direction (within an acute angle) are positively correlated. Generally, higher values of lamina wax, petiole wax, and NPCI were associated with the stress environments, whereas higher values of flowering duration, stem thickness, and NDVI were associated with the control condition.

### 3.2. Genome-Wide Association Analysis

For the association analysis, a total of 16,877 previously identified SNPs by GBS were used to determine marker–trait associations [29]. Association analysis detected significant SNP markers for averaged values of the traits across all environments using the multi-locus mixed-model (MLMM) analysis based on BLUP values, as well as for individual environments for each trait. Association mapping for lamina wax concentration identified four SNPs (Chr1LG6_277526227, Chr4LG4_209093982, Chr6LG2_384797968, and Chr7LG7_128419954) at a significant level of −Log10 (*p*) > 5.5; *p* ≤ 3.7 × 10^−6^ (Table 3; Figure 3A). Three SNPs (Chr4LG4_16602920, Chr7LG7_346970562, and Uscaffold03717_87257 were detected for petiole wax (Table 3; Figure 3B). For both lamina and petiole waxes, the detected SNPs were also significant in at least two of the three environments (Table 3).

For stem thickness, association mapping identified three SNPs (Chr7LG7_120991008, Chr7LG7_415249611, and Uscaffold03985_59708) significantly (−Log10 (*p*) > 5; *p*
*≤* 3.20 × 10^−6^) associated with the average stem thickness across five environments (Table 3; Figure 3C). The SNPs included two on chromosome 7 (LG7) and one on a non-chromosomal scaffold. The detected SNPs were significant in four of the five environments.

Two SNPs (Chr3LG5_18677470 and Chr5LG3_255645703) were significantly (−Log10 (*p*) > 3.5; *p* ≤ 5.9 × 10^−4^) associated with the flowering duration across the mean of five environments. The detected SNPs were also significant in at least in three of the five environments.

One SNP (Chr6LG2_21764881) on chromosome 6 (LG2) was associated with NDVI at a significance level of (−Log10 (*p*) > 4; *p* ≤ 1.3 × 10^−4^) (Table 3; Figure 3E). Finally, two SNPs (Chr5LG3_566189589 and Chr6LG2_464876174) were identified for NPCI at a significance level of (−Log10 (*p*) > 4.5; *p* ≤ 4.9 × 10^−4^). For all traits phenotyped at five environments, the detected SNPs also had a significant marker–trait association in at least three of the five environments. For all traits, Manhattan plots depicting the distribution of SNP markers along with the *p*-value of each MTAs and the corresponding Q-Q plots are presented in Figure 3A–F. 

The 15 significant SNPs detected from the association analysis were used to identify candidate genes for the six traits. Sixteen unique genes were identified within a 15 kb region on either side of the significant SNP markers and are considered potential causative candidate genes (Table 4). The biological processes associated with these candidate genes include the biotin biosynthetic process, actin filament polymerization, and protein autophosphorylation. The molecular and cellular functions of the candidate genes are listed in Table 4. The GO term GO:0016021 identified as an integral component of membranes is associated with candidate genes associated with all six phenotypes measured in this study. 

## 4. Discussion

Environmental stresses, mainly heat and drought, cause substantial yield loss in pea [3,4]. Due to climate change, high air temperature, extreme heat events, and the increasing frequency and intensity of drought are impeding crop production and yield in pea-growing regions across the globe. In Saskatchewan, Canada, the largest producer and exporter of pea in the world, the 2021 field season was the most drought- and heat-stressed in the past 50 years, causing an approximately 37% decrease in pea yield compared to the five-year average (https://agriculture.canada.ca (accessed on 14 October 2021)). This climate scenario underlines the need to select cultivars that are able to grow and yield well under stress conditions. The selection of stress-resistant cultivars should primarily rely on the use of stress-adaptive traits or related proxies such as vegetation indices. Pea has substantial genetic diversity, and strategic use of the available variation through using allelic variation is crucial to enhance stress resistance. With the presence of affordable and cutting-edge SNP genotyping technology and vast genomic resources, GWAS has been a dependable method for detecting genetic regions associated with traits of interest in several crops [29,40,41,42]. 

GWAS previously conducted on pea have identified several markers and candidate genes associated with agronomic and seed quality traits, disease resistance, root architecture, essential minerals [29,33,34,43], and other traits of agronomic interest. In our study, we conducted a GWAS to identify SNP markers associated with six heat and drought adaptive traits (lamina wax, petiole wax, stem thickness, flowering duration, NDVI, and NPCI) using 135 genetically diverse pea accessions. 

All traits accounted in this study were significantly affected by both genotype and environment effects. We observed moderate to high broad-sense heritability values for the traits, which is in agreement with previous reports on pea in related traits [35,41,44]. Overall, association analysis identified 15 SNPs significantly associated with stress-adaptive traits and the markers were distributed over six of the seven chromosomes and a non-chromosomal scaffold (Table 3). Gali et al. [29] indicated that a significant marker detected for a given trait would be more trustworthy if the marker has great reproducibility and would be found in multiple trials. Therefore, for the six traits we examined, the SNP markers declared significant were consistent in at least two of the three environments for lamina and petiole wax, and three of the five environments for the remaining four traits. The detected markers could be used for marker-assisted selection of these traits in the breeding effort of developing stress-resistant pea cultivars. 

A total of four SNPs (Chr1LG6_277526227, Chr4LG4_209093982, Chr6LG2_384797968, and Chr7LG7_128419954) for lamina wax, and three SNPs (Chr4LG4_16602920, Chr7LG7_346970562, and Uscaffold03717_87257) for petiole wax were detected on different chromosomes and the non-chromosomal scaffold. We believe that this is the first report evaluating waxes in pea by GWAS. Greater lamina and petiole wax concentrations were associated with the stress environment, which is in agreement with several studies conducted on different crops species [14,16,45,46]. Abiotic stresses triggered increased wax concentrations as a stress-resistance response [12,47]. For example, as a drought avoidance mechanism in pea, epicuticular wax reduced residual transpiration and thus minimized water loss so that tissue water status was maintained [14]. Tafesse [18] reported that the leaf surface wax concentration was positively correlated with the water band index, an indication of high leaf water potential, and contributed to a cooler canopy. Similarly, as a heat avoidance mechanism, epicuticular wax protects leaves and stems from powerful radiation that would impose radiation stress and from excess heat by reflecting ultraviolet, visible, and infrared wavelengths [12,18]. Our results show heat induced more wax in stressed environments, although part of the thicker wax deposition could be explained by a concomitant reduction in leaf size in stress because leaf expansion is a process sensitive to stress. 

Epicuticular wax is a quantitative trait controlled by several genes involved in the biosynthesis and transport of wax to the outer membrane in plants, and the expression of these genes is highly dependent on the environment. Xue et al. [12] reported that several genes involved in wax biosynthesis were upregulated under stress conditions. WAX2 is among the genes responsible for wax formation in Arabidopsis [48], and glossy13 has a similar role in maize [49]. ABC transporters are necessary for wax export from the site of synthesis to the outer membrane in plants [50]. Based on a GWAS study conducted on sorghum, Elango et al. [16] reported several putative genes responsible for epicuticular-wax biosynthesis and export from the synthesis site to the outer plant membrane. Selecting genotypes for high epicuticular wax is essential in breeding stress-resistant crop cultivars for increased adaptation to environmental stresses. 

Three SNPs, two of them (Chr7LG7_120991008, Chr7LG7_415249611) on chromosome 7 (LG7) and one (Uscaffold03985_59708) on the non-chromosomal scaffold, were detected with a significant association with stem thickness. In a study of a pea bi-parental mapping population, Gawłowska et al. [51] reported several QTLs associated with stem mechanical properties including stem diameter and wall thickness, traits strongly relate to lodging resistance. Both genotype and environment had significant effects on stem thickness. Stem thickness ranged from 2.42 mm to 4.81 mm in the 135 accessions and 5 environments. Heat and drought stress reduced the stem thickness by 15%. In addition to lodging, stem thickness was also reported to be associated with disease resistance and seed yield [52,53]. Le, also known as Mendel’s tall/dwarf gene, controls plant height and stem diameter [54]. Stem thickness enhances heat and drought resistance directly by maintaining water in the stem and improving leaf water potential, which stabilizes crop yield under heat and drought stresses [17,55]. Stem thickness also contributes to stress resistance indirectly by improving stem strength and making the plant more upright and resistant to lodging. From our recent publication by Tafesse et al. [4], we have strong evidence that lodging-resistant upright cultivars have greater resistance to heat and drought stresses. Similarly, Smitchger et al. [53] indicated that stem diameter had a strong correlation with lodging resistance. Lodging is strongly associated with high canopy temperature, indicating greater heat and drought stresses [4]. A possible elucidation for the contribution of lodging on enhancing canopy temperature is that lodged plants in the field make direct contact with the sun-heated soil surface, from which heat can transfer to the plant canopy as conducted heat. Moreover, lodged canopies are planophile, and on hot days they absorb more heat from the sun and reflect less at near-infrared wavelengths, and this further causes a hotter canopy [4]. As a drought-resistance trait, thicker stems conduct and maintain more water in plant tissues and contribute to greater water use efficiency. Tafesse [18] reported that stem thickness positively correlates with the water band index, a proxy for plant water content. Therefore, breeding efforts to improve pea stress tolerance should focus on thicker and stronger stems. 

Both heat and drought stress significantly shorten the flowering duration, which means less time for flower development and pod formation, and thus lower grain yield. Flowering duration is a quantitative trait under complex genetic and environmental control [41]. Here we detected two SNPs (Chr3LG5_18677470; Chr5LG3_255645703) associated with flowering duration. Previously, Jiang et al. [41] and Huang et al. [44] reported QTL for the flowering duration on LG3, and one of the two SNPs we identified in this study is also in LG3. Flowering duration and reproductive nodes have strong positive correlations, and several QTL were reported for reproductive nodes on LG III [44,56] Reproductive node number significantly correlates with pod number and seed yield in pea [4]. The broad-sense heritability of flowering duration was 0.82; previously Jiang et al. [41] reported 0.78. Environmental variables, particularly temperature and moisture, have major impacts on the flowering duration [3,57]. In a controlled environment study, Tafesse [18] reported the combined occurrence of heat and drought reduced the flowering duration by 33%. In the present study, the flowering duration ranged from 13 to 39 days across 135 accessions and 5 environments. The 2015 Saskatoon and 2016 Rosthern environments had the shortest (20 days) and longest (29 days) mean flowering durations, respectively. Indeterminacy and a longer flowering duration buffered grain yield in pea under environmental stresses [44]. Breeding efforts to improve seed yield in pea should focus on selecting cultivars with a longer flowering duration, which means more reproductive nodes and pods to compensate for lost yield under stress conditions.

One locus (Chr6LG2_21764881) was associated with NDVI. NDVI has long been recognized as a reliable proxy for effectively estimating crop biomass, canopy greenness, and grain yield [22,24,58]. NDVI is responsive to both genotype and environmental variables, and in this study its value diminished under the stress environments. The broad-sense heritability was 0.70. There are only limited studies that have applied GWAS to detect markers associated with spectral indices. Previously, we reported two markers associated with the photochemical reflectance index (PRI), a proxy of crop photosynthetic efficiency and the xanthophyll cycle, that respond to environmental variables [35]. Several QTLs were associated with NDVI in wheat [59]. With the advancement of remote-sensing-based phenotyping platforms, breeders are adopting spectral technologies to be used as a proxy to evaluate plant growth and yield performance, disease resistance, and abiotic stress effects [23,24,25,60]. Similarly, two SNPs (Chr5LG3_566189589 and Chr6LG2_464876174) were associated with the normalized pigment and chlorophyll index (NPCI). In a GWAS study conducted on wheat, Gizaw et al. [26] reported certain markers associated with NPCI. Again, as far as we know, no GWAS study has previously been employed using NPCI as a trait of stress resistance in pea. NPCI effectively estimates chlorophyll and pigment degradation, and a higher value of NPCI is associated with less pigment absorption in the red relative to the blue region in the light spectrum [21]. The principal component analysis (Figure 2) clearly shows high NPCI value associated with stress environments. Based on our results and others in the literature, vegetation indices including NDVI and NPCI can be effective proxies for plant vigor, yield, and stress resistance, to be adopted as selection criteria in breeding programs that aim to enhance yield and stress resistance. The GO term GO:0016021 is associated with potential candidate genes identified for lamina wax, petiole wax, stem thickness, flowering duration as well those identified for NDVI and NPCI. This common annotation of candidate genes may be indirect support for the effectiveness of NDVI and NPCI. 

In conclusion, in the present GWAS, we observed significant phenotypic variation in six stress-adaptive traits (lamina wax, petiole wax, stem thickness, flowering duration, normalized difference vegetation index (NDVI), and normalized pigment and chlorophyll index (NPCI)) among 135 pea accessions in multi-environment tests. Further, we identified 15 SNPs significantly (*p* ≤ 0.0005) associated with the six stress-adaptive traits. These results are believed to advance the knowledge of the genetic bases governing these traits. The detected SNPs should be useful for marker-assisted selection for breeders in developing stress-resistant pea cultivars.

## Figures and Tables

**Figure 1 genes-12-01897-f001:**
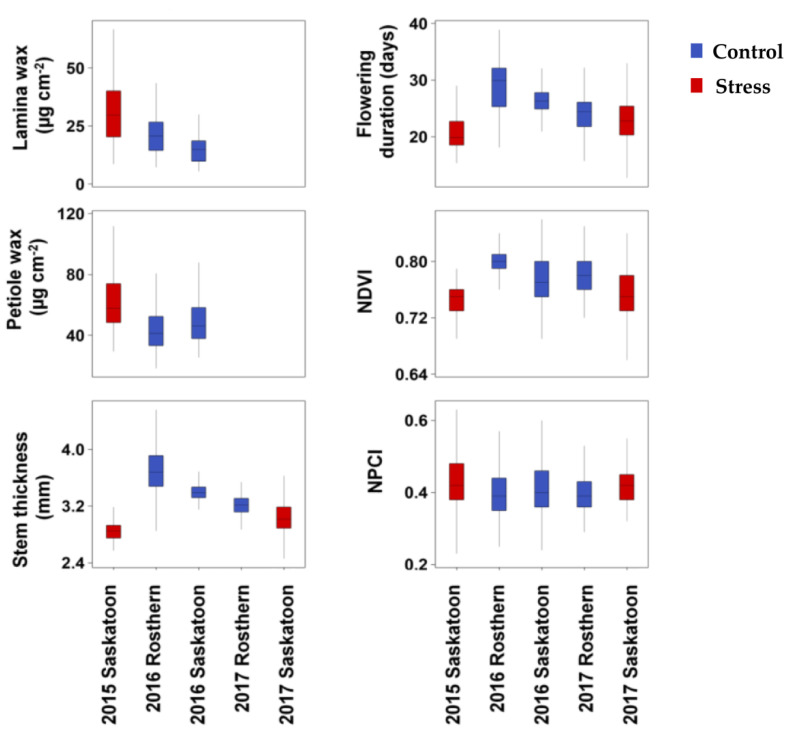
Box plots of lamina wax, petiole wax, stem thickness, flowering duration, normalized difference vegetation index (NDVI), and normalized pigment and chlorophyll index (NPCI) of 135 pea accessions grown under control and stress conditions across multiple environments in Saskatchewan, Canada. Note: The control environments were 2016 Rosthern, 2016 Saskatoon, and 2017 Rosthern; the stress environments were 2015 and 2017 Saskatoon.

**Figure 2 genes-12-01897-f002:**
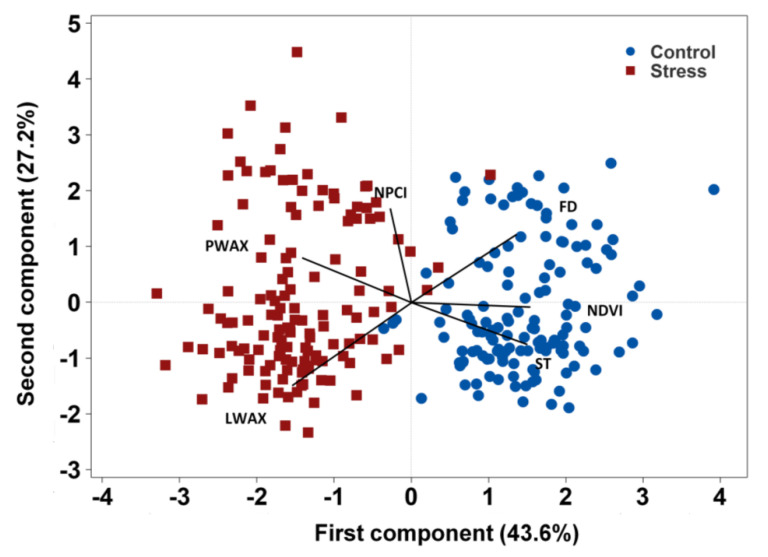
Bi-plot of principal component analysis depicting the overall traits association and accessions response to the environment. Note: The control environments were 2016 Rosthern, 2016 Saskatoon, and 2017 Rosthern; the stress environments were 2015 and 2017 Saskatoon. LWAX, lamina wax; PWAX, petiole wax; ST, stem thickness, FD, flowering duration; NDVI, normalized difference vegetation index, and NPCI, normalized pigment and chlorophyll index.

**Figure 3 genes-12-01897-f003:**
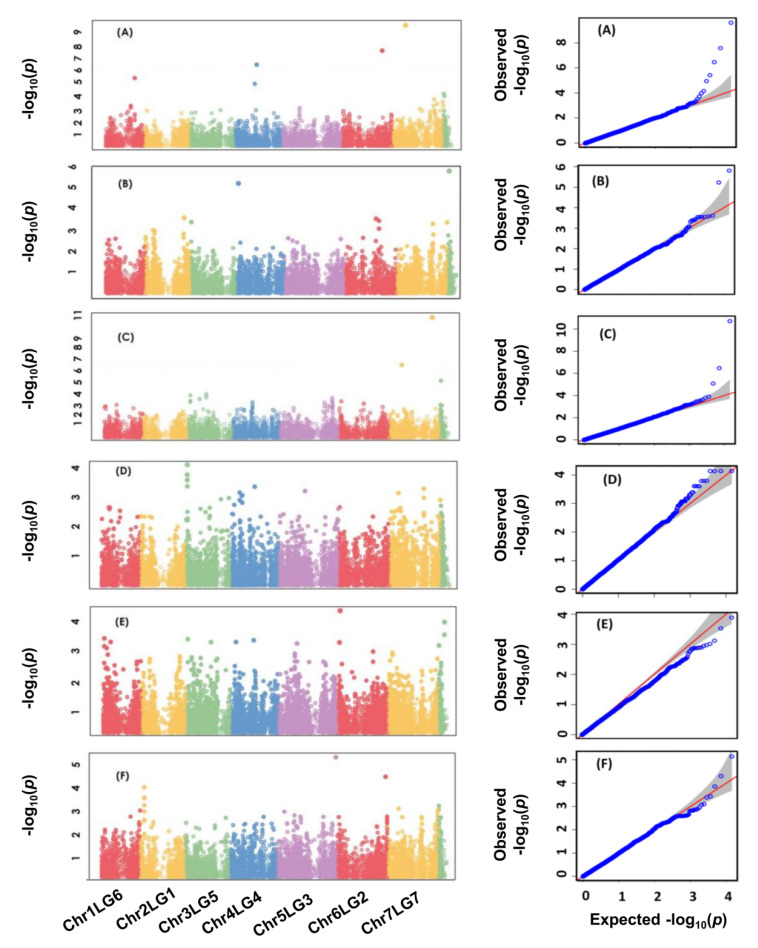
Manhattan plots and the corresponding Q-Q plots displaying the *p* values of the identified SNP markers to be associated with lamina wax (**A**), petiole wax (**B**), stem thickness (**C**), flowering duration (**D**), normalized difference vegetation index (**E**), and normalized pigment and chlorophyll index (**F**). The Manhattan plots are determined using a total of 16,877 SNP markers of 135 pea accessions in the multi-environment experiments.

**Table 1 genes-12-01897-t001:** Variance components of genotype, environment, G × E interaction, and the broad-sense heritability (*H*^2^) of lamina wax, petiole wax, stem thickness, flowering duration, NDVI, and NPCI of 135 pea accessions grown across three or five environments (2015 Saskatoon, 2016 Rosthern, 2016 Saskatoon, 2017 Rosthern, and 2017 Saskatoon).

Source	Lamina Wax		Petiole Wax		Stem Thickness		Flowering Duration		NDVI		NPCI	
	Variance	% of Total	Variance	% of Total	Variance	% of Total	Variance	% of Total	Variance	% of Total	Variance	% of Total
Genotype (G)	55.5 ***	36.0%	101.1 ***	39.8%	0.059 ***	34.7%	8.7 ***	36.4%	0.00054 ***	45.0%	0.0028 ***	61.4%
Environment(E)	64.0 ***	41.5%	92.2 ***	36.3%	0.058 ***	33.9%	10.6 ***	44.6%	0.0002 ***	16.7%	0.00014 ***	3.0%
Replication	2.2 **	1.4%	9.7 **	3.8%	0.006 *	3.3%	0.0 ns	0.0%	0.0001 *	0.0%	0.000034 ***	0.7%
G × E	0.0 ns	0.0%	5.1 ns	2.0%	0.002 ns	1.2%	1.6 ***	6.6%	0.0001	0.0%	0.0 ns	0.0%
Error	32.1	21.1%	71.4	28.1%	0.046	26.9%	3.0	12.5%	0.00046	38.4%	0.0016	34.9%
Total	154.3		279.5		0.171		23.8		0.0012		0.0046	
(H^2^)	0.78		0.73		0.72		0.82		0.70		0.78	

*, **, *** indicates significance at the 0.05, 0.01, and 0.001 probability levels, respectively; ns indicates not significant at the 0.05 level. NDVI, normalized difference vegetation index; NPCI, normalized pigment and chlorophyll index.

**Table 2 genes-12-01897-t002:** Descriptive statistics for minimum, maximum, mean, and standard deviation of lamina wax, petiole wax, stem thickness, flowering duration, normalized difference vegetation index (NDVI), and normalized pigment and chlorophyll index (NPCI) of 135 pea accessions of the genome-wide association study panel evaluated at multiple locations in Saskatchewan, Canada.

Trait	Environment	Minimum	Maximum	Mean	Standard Deviation
Lamina wax (µg cm^−2^)	2015 Saskatoon	8.6	66.8	30.9	12.9
	2016 Rosthern	7.2	43.5	21.1	7.9
	2016 Saskatoon	5.4	33.4	15.0	6.0
Petiole wax (µg cm^−2^)	2015 Saskatoon	29.4	140.1	63.2	23.3
	2016 Rosthern	18.2	110.9	45.4	17.6
	2016 Saskatoon	25.3	114.5	49.7	16.8
Stem thickness (mm)	2015 Saskatoon	2.57	3.25	2.85	0.13
	2016 Rosthern	2.76	4.81	3.70	0.37
	2016 Saskatoon	3.06	3.80	3.40	0.12
	2017 Rosthern	2.87	3.64	3.22	0.13
	2017 Saskatoon	2.42	3.70	3.03	0.23
Flowering duration (days)	2015 Saskatoon	15.3	29.0	20.6	2.9
	2016 Rosthern	18.1	38.9	29.0	4.5
	2016 Saskatoon	17.5	35.6	26.6	3.0
	2017 Rosthern	14.8	36.6	24.4	3.7
	2017 Saskatoon	12.7	33.0	22.9	3.8
NDVI	2015 Saskatoon	0.70	0.80	0.76	0.02
	2016 Rosthern	0.74	0.85	0.79	0.02
	2016 Saskatoon	0.63	0.85	0.76	0.04
	2017 Rosthern	0.71	0.84	0.77	0.03
	2017 Saskatoon	0.64	0.85	0.77	0.03
NPCI	2015 Saskatoon	0.23	0.69	0.43	0.07
	2016 Rosthern	0.21	0.62	0.41	0.06
	2016 Saskatoon	0.25	0.70	0.42	0.07
	2017 Rosthern	0.22	0.57	0.40	0.06
	2017 Saskatoon	0.21	0.65	0.41	0.06

**Table 3 genes-12-01897-t003:** Significant SNP markers detected to be associated with six traits. The markers were identified for the six traits by association analysis of 135 pea accessions evaluated in three environments for lamina and petiole waxes, and five environments for the remaining four traits in Saskatchewan, Canada.

Trait	SNP	Environment	*p* Value	MAF
Lamina wax	Chr1LG6_277526227	2015 Saskatoon	1.30 × 10^−4^	0.08
		2016 Rosthern	2.90 × 10^−3^	0.08
		2016 Saskatoon	6.20 × 10^−4^	0.08
		BLUPs	3.70 × 10^−6^	0.08
	Chr4LG4_209093982	2015 Saskatoon	8.00 × 10^−3^	0.11
		2016 Rosthern	2.10 × 10^−3^	0.11
		2016 Saskatoon	2.50 × 10^−3^	0.11
		BLUPs	3.30 × 10^−7^	0.11
	Chr6LG2_384797968	2015 Saskatoon	4.30 × 10^−5^	0.47
		2016 Rosthern	3.10 × 10^−4^	0.47
		2016 Saskatoon	1.50 × 10^−6^	0.47
		BLUPs	2.50 × 10^−8^	0.47
	Chr7LG7_128419954	2015 Saskatoon	3.20 × 10^−6^	0.32
		2016 Rosthern	8.90 × 10^−4^	0.32
		2016 Saskatoon	1.10 × 10^−6^	0.32
		BLUPs	2.50 × 10^−10^	0.32
Petiole wax	Chr4LG4_16602920	2015 Saskatoon	2.40 × 10^−2^	0.36
		2016 Rosthern	5.20 × 10^−2^	0.37
		2016 Saskatoon	2.52 × 10^−2^	0.37
		BLUPs	5.80 × 10^−6^	0.37
	Chr7LG7_346970562	2015 Saskatoon	1.10 × 10^−9^	0.12
		2016 Rosthern	2.40 × 10^−2^	0.12
		2016 Saskatoon	2.72 × 10^−2^	0.12
		BLUPs	4.80 × 10^−4^	0.12
	Uscaffold03717_87257	2015 Saskatoon	7.20 × 10^−3^	0.42
		2016 Saskatoon	9.90 × 10^−3^	0.42
		BLUPs	1.50 × 10^−6^	0.42
Stem thickness (mm)	Chr7LG7_120991008	2015 Saskatoon	2.00 × 10^−4^	0.31
		2016 Saskatoon	6.70 × 10^−4^	0.31
		2017 Rosthern	3.00 × 10^−3^	0.31
		2017 Saskatoon	4.20 × 10^−3^	0.31
		BLUPs	3.20 × 10^−7^	0.31
	Chr7LG7_415249611	2015 Saskatoon	1.30 × 10^−5^	0.18
		2016 Rosthern	4.20 × 10^−8^	0.18
		2016 Saskatoon	7.80 × 10^−9^	0.18
		2017 Rosthern	6.10 × 10^−8^	0.18
		BLUPs	1.90 × 10^−11^	0.18
	Uscaffold03985_59708	2015 Saskatoon	3.60 × 10^−4^	0.28
		2016 Rosthern	5.00 × 10^−5^	0.28
		2016 Saskatoon	8.50 × 10^−4^	0.28
		2017 Saskatoon	6.40 × 10^−3^	0.28
		BLUPs	8.30 × 10^−6^	0.28
Flowering duration (days)	Chr3LG5_18677470	2015 Saskatoon	3.20 × 10^−6^	0.18
		2016 Rosthern	4.40 × 10^−4^	0.18
		2017 Saskatoon	1.60 × 10^−5^	0.18
		BLUPs	7.30 × 10^−5^	0.18
	Chr5LG3_255645703	2015 Saskatoon	6.80 × 10^−5^	0.17
		2016 Saskatoon	7.30 × 10^−3^	0.17
		2017 Rosthern	2.70 × 10^−4^	0.17
		2017 Saskatoon	2.20 × 10^−8^	0.17
		BLUPs	5.90 × 10^−4^	0.17
NDVI	Chr6LG2_21764881	2016 Saskatoon	8.60 × 10^−3^	0.09
		2017 Rosthern	1.40 × 10^−4^	0.09
		2017 Saskatoon	3.90 × 10^−5^	0.09
		BLUPs	1.30 × 10^−4^	0.09
NPCI	Chr5LG3_566189589	2015 Saskatoon	9.80 × 10^−4^	0.36
		2016 Saskatoon	8.90 × 10^−3^	0.36
		2017 Rosthern	4.30 × 10^−3^	0.36
		2017 Saskatoon	6.60 × 10^−5^	0.36
		BLUPs	7.10 × 10^−6^	0.36
	Chr6LG2_464876174	2015 Saskatoon	2.80 × 10^−3^	0.30
		2016 Rosthern	5.60 × 10^−3^	0.30
		2016 Saskatoon	1.70 × 10^−2^	0.30
		2017 Rosthern	5.00 × 10^−3^	0.30
		2017 Saskatoon	1.70 × 10^−3^	0.30
		BLUPs	4.90 × 10^−5^	0.30

Note: All the significant markers reported here were also significant in at least two of the three environments for lamina and petiole waxes, and three of the five environments for the remaining four traits. In each SNP name, Chr indicates chromosome and LG indicates linkage group and the numbers followed after the dash indicate the base pair position. Sc refers to scaffold for non-chromosomal SNPs followed by the respective scaffold number. Each locus is represented by one SNP marker of the LD block [37]. MAF, minor allele frequency.

**Table 4 genes-12-01897-t004:** Candidate genes that are detected within 15 kb distance on either side of the SNP markers identified for association with the six stress-adaptive traits in pea.

Trait	SNP Marker	Gene ID	Protein Name	Gene_Name	Organism	Gene Ontology IDs	Molecular Function	Cellular Component
Lamina wax	Chr1LG6_277526227	Psat1g139360	Hydrolase activity + hydrolyzing O-glycosyl compounds	D0Y65_006627	*Glycine soja*			
	Chr4LG4_209093982	Psat4g112480	Arp2/3 complex + 34 kD subunit p34-Arc	11418544 MTR_8g070640	*Medicago truncatula*	GO:0005885; GO:0005737; GO:0051015; GO:0005200; GO:0030041; GO:0034314	actin filament binding [GO:0051015]; structural constituent of cytoskeleton [GO:0005200]	Arp2/3 protein complex [GO:0005885];cytoplasm [GO:0005737]
	Chr7LG7_128419954	Psat7g076840	NnrU protein	11437558 MTR_8g097190 MtrunA17_Chr8g0377611	*Medicago truncatula*	GO:0016021; GO:0016853	isomerase activity [GO:0016853]	integral component of membrane [GO:0016021]
Petiole wax	Chr4LG4_16602920	Psat4g011120	Aminotransferase class-III	11446047 MTR_4g128620 MtrunA17_Chr4g0072721	*Medicago truncatula*	GO:0005739; GO:0004015; GO:0004141; GO:0030170; GO:0009102	adenosylmethionine-8-amino-7-oxononanoate transaminase activity [GO:0004015]; dethiobiotin synthase activity [GO:0004141]; pyridoxal phosphate binding [GO:0030170]	mitochondrion [GO:0005739]
	Chr7LG7_346970562	Psat7g186040	Pyridine nucleotide-disulphide oxidoreductase	LOC101505252	*Cicer arietinum*	GO:0005739; GO:0016491	oxidoreductase activity [GO:0016491]	mitochondrion [GO:0005739]
	Sc03717_87257	Psat0s3717g0080	Unknown gene	LOC101501731	*Cicer arietinum*	GO:0016021		integral component of membrane [GO:0016021]
Stem thickness	Chr7LG7_120991008	Psat7g071920	Unknown gene	L195_g021419	*Trifolium pratense*	GO:0005634		nucleus [GO:0005634]
	Chr7LG7_120991008	Psat7g072040	Protein of unknown function (DUF616)	11413795 MTR_8g085850 MtrunA17_Chr8g0378731	*Medicago truncatula*	GO:0016021		integral component of membrane [GO:0016021]
	Chr7LG7_415249611	Psat7g208760	Unknown gene	TSUD_89070	*Trifolium subterraneum*			
	Sc03985_59708	Psat0s3985g0040	Myb/SANT-like DNA-binding domain	MtrunA17_Chr1g0176881	*Medicago truncatula*			
Flowering duration	Chr3LG5_18677470	Psat3g006600	Protein of unknown function (DUF3353)	MtrunA17_Chr7g0274601	*Medicago truncatula*	GO:0016021		integral component of membrane [GO:0016021]
	Chr5LG3_255645703	Psat5g140600	SWIB/MDM2 domain	TSUD_394050	*Trifolium subterraneum*			
NDVI	Chr6LG2_21764881	Psat6g028080	PB1 domain	FH972_013116	*Carpinus fangiana*	GO:0005509	calcium ion binding [GO:0005509]	
	Chr6LG2_21764881	Psat6g028120	Protein kinase domain	11426285 MTR_5g024450 MtrunA17_Chr5g0407241	*Medicago truncatula*	GO:0016021; GO:0005886; GO:0005524; GO:0004674; GO:0046777	ATP binding [GO:0005524]; protein serine/threonine kinase activity [GO:0004674]	integral component of membrane [GO:0016021];plasma membrane [GO:0005886]
NPCI	Chr5LG3_566189589	Psat5g299040	PPR repeat family	LOC101504534	*Cicer arietinum*			
	Chr6LG2_464876174	Psat6g231000	Dual specificity phosphatase + catalytic domain	25485578 MTR_1g112080 MtrunA17_Chr1g0210681	*Medicago truncatula*	GO:0016021; GO:0008138	protein tyrosine/serine/threonine phosphatase activity [GO:0008138]	integral component of membrane [GO:0016021]

## Data Availability

This manuscript includes the essential data used either as tables or figures in the result section. The sequence information of the SNP markers is available at (https://www.ebi.ac.uk/ena/browser/view/PRJEB35147, accessed on 15 April 2021). Weather data are available in the Environment Canada database at (https://climate.weather.gc.ca, accessed on 12 January 2020).

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
