# Peer review of "Genome-Wide Association Mapping for Heat and Drought Adaptive Traits in Pea"

_genes, 2021, doi:10.3390/genes12121897_

Round 1

Reviewer 1 Report

This paper conducted a GWAS study in pea on physiological and agronomic traits. 15 marker-trait associations (MTAs) were detected and sixteen candidate genes were identified, providing helpful information for molecular breeding stress resistant cultivars in pea. The results are new and reliable, the writing organization is reasonable, I suggest acceptance of the manuscript.

Author Response

We appreciate the careful reviews which have helped to improve the manuscript. Following are our detailed responses to the reviewers’ suggestions. Our responses are preceded by ‘TW reply’.   

Reviewer 1

This paper conducted a GWAS study in pea on physiological and agronomic traits. 15 marker-trait associations (MTAs) were detected and sixteen candidate genes were identified, providing helpful information for molecular breeding stress resistant cultivars in pea. The results are new and reliable, the writing organization is reasonable, I suggest acceptance of the manuscript.

TW reply: We appreciate the generous comments of Reviewer 1 and thank you for recommending acceptance of our paper for publication. 

Reviewer 2 Report

This is a well-written paper highly interesting to pea and legume researchers and breeders. I've only few comments which are focussed on the difference between stress-adaptive traits vs. stress resistance (or tolerance):

General:

Although the title of the paper correctly addresses "adaptive traits" rather than stress tolerance (as the ultimate trait relevant to breeders and farmers) itself, it should be outlined more clearly in the text (esp. Discussion) that heat and drought resistance have not been assessed per se, e. g., by comparing yields under stress vs. control conditions.

For instance, in line 76 the notion "...to explore the genetic variation of stress tolerance..." is potentially misleading in this sense.

dt. line 363 "...significantly associated with stress tolerance..."

l. 161

Formula should better we written using superscripts.

l. 172

remove full stop

l. 406 - 412

"Both genotype and environment had significant effects on stem thickness.": O.k.,  but how about the effect of stem thickness itself on heat and drought-stress tolerance? Is it correct to conclude that if stress effects stem thickness, then stem thickness would itself have an effect on stress tolerance?

"Here we report that stem thickness contributes to heat and drought resistance."

No data presented for this clear-cut notion. Rather, this is an interpretation (indirect conclusion) of the the results.

"Stem thickness enhances heat resistance indirectly..."

O.k., however that's a knowledge cited from other studies and is not a result of the present study. The cited knowledge may, of course, been used when trying to make some sense of the present study's data. However it should be clearly stated that this sense represents an interpretation rather than a finding.

Author Response

We appreciate the careful reviews which have helped to improve the manuscript. Following are our detailed responses to the reviewers’ suggestions. Our responses are preceded by ‘TW reply’.   

Reviewer 2

This is a well-written paper highly interesting to pea and legume researchers and breeders. I've only few comments which are focused on the difference between stress-adaptive traits vs. stress resistance (or tolerance):

General:

Although the title of the paper correctly addresses "adaptive traits" rather than stress tolerance (as the ultimate trait relevant to breeders and farmers) itself, it should be outlined more clearly in the text (esp. Discussion) that heat and drought resistance have not been assessed per se, e. g., by comparing yields under stress vs. control conditions.

For instance, in line 76 the notion "...to explore the genetic variation of stress tolerance..." is potentially misleading in this sense.

TW reply: ‘Stress tolerance’ has been replaced with ‘stress adaptive traits’ and the sentence was revised as: ‘… to explore the genetic variation of stress adaptive traits present in a GWAS panel of 135 accessions …’

  1. line 363 "...significantly associated with stress tolerance..."

TW reply: ‘Stress tolerance’ has been replaced with ‘stress adaptive traits’ and the sentence was revised as: ‘Overall, association analysis identified 15 SNPs significantly associated with stress adaptive traits and the markers were distributed over six of the seven chromosomes, and a non-chromosomal scaffold’.

  1. 161

Formula should better we written using superscripts.

TW reply: The formula has been revised as recommended.  

  1. 172

remove full stop

TW Reply: full stop has been removed, thank you for noticing this.

  1. 406 - 412

"Both genotype and environment had significant effects on stem thickness.": O.k.,  but how about the effect of stem thickness itself on heat and drought-stress tolerance? Is it correct to conclude that if stress effects stem thickness, then stem thickness would itself have an effect on stress tolerance?

"Here we report that stem thickness contributes to heat and drought resistance."

No data presented for this clear-cut notion. Rather, this is an interpretation (indirect conclusion) of the results.

TW Reply: This is a very good point and we agree that the sentence “Here we report that stem thickness contributes to heat and drought resistance” can be misleading and the sentence has been removed and replaced with a more detailed explanation as described below.

Although in this study we do not provide data on how stem thickness contributes directly to heat and drought stress tolerance, as this was not our main goal, there is a body of evidences that shows that stem thickness contributes to both heat and drought tolerance.  The main point we are making here is that thicker stems contribute to stress resistance directly by maintaining  water in the stem which improves leaf water potential, and thus contributes to stabilizing yield under heat and drought stress (Klepper, 1971; Sallam et al., 2014; Tafesse, 2018).  Stem thickness also contribute to stress resistance indirectly by improving stem strength and making the pea plant upright and resistant to lodging. From our recent publication (Tafesse et al., 2019), we have strong evidence that lodging resistant upright cultivars have greater resistance to heat and drought stresses. We believe this concept is reflected in the current manuscript and our conclusions are derived from these data.

"Stem thickness enhances heat resistance indirectly..."

TW Reply: This sentence has been removed and replaced with a better explanation.

O.k., however that's a knowledge cited from other studies and is not a result of the present study. The cited knowledge may, of course, been used when trying to make some sense of the present study's data. However it should be clearly stated that this sense represents an interpretation rather than a finding.

TW Reply: This comment is accepted and the associated sentences are improved.